# CascadedGaze: Efficiency in Global Context Extraction for Image Restoration

**Amirhosein Ghasemabadi**[1,2]**, Muhammad Kamran Janjua**[*2]**, Mohammad Salameh**[*2]**,
Chunhua Zhou**[3]**, Fengyu Sun**[3]**, Di Niu**[1]

[1]*Dept. ECE, University of Alberta, Canada,* [2]*Huawei Technologies, Canada,* [3]*Huawei Kirin Solution, China*
*{ghasemab,dniu}@ualberta.ca, {kamran.janjua, mohammad.salameh, zhouchunhua}@huawei.com,*
*sunfengyu@hisilicon.com*
[*]Equal Contribution

**Reviewed on OpenReview:** *https: // openreview. net/ forum? id=C3FXHxMVuq*

## Abstract

Image restoration tasks traditionally rely on convolutional neural networks. However, given the local nature of the convolutional operator, they struggle to capture global information. The promise of attention mechanisms in Transformers is to circumvent this problem, but it comes at the cost of intensive computational overhead. Many recent studies in image restoration have focused on solving the challenge of balancing performance and computational cost via Transformer variants. In this paper, we present CascadedGaze Network (CGNet), an encoder-decoder architecture that employs Global Context Extractor (GCE), a novel and efficient way to capture global information for image restoration. The GCE module leverages small kernels across convolutional layers to learn global dependencies, without requiring self-attention. Extensive experimental results show that our computationally efficient approach performs competitively to a range of state-of-the-art methods on synthetic image denoising and single image deblurring tasks, and pushes the performance boundary further on the real image denoising task. We release the code at the following link: `https://github.com/Ascend-Research/CascadedGaze`.

## 1 Introduction

Image restoration refers to recovering the original image quality by addressing degradation introduced during the capture, transmission, and storage processes. This degradation includes unwanted elements like noise, blurring, and artifacts. Given that infinitely many feasible solutions may exist, image restoration is considered an ill-posed problem. It is a challenging task as it involves processing high-frequency elements like noise while preserving crucial image characteristics such as edges and textures (Su et al., 2022b). To tackle this complexity, current image restoration techniques leverage deep neural networks. These networks have demonstrated remarkable progress across various restoration tasks, achieving state-of-the-art results on several benchmark datasets (Li et al., 2023b; Zamir et al., 2021; Wang et al., 2022b; Cheng et al., 2021; Chu et al., 2022).

While convolutional neural networks (CNNs) have been widely used for image restoration (Chen et al., 2021; Fan et al., 2022; Chang et al., 2020; Yue et al., 2020), their limited receptive field size restricts their ability to capture long-range dependencies and global context effectively. Conversely, Transformers excel at modeling global interactions and dependencies, making them well-suited for image restoration tasks that require a holistic understanding of the image content (Dosovitskiy et al., 2020; Vaswani et al., 2017; Ramachandran et al., 2019; Touvron et al., 2021). However, Transformers come at the cost of intensive memory consumption and quadratic computational complexity of self-attention as image spatial resolution increases.

Due to the computational overhead of Transformers, especially self-attention, there has been a growing interest in developing efficient types of Transformers. Various techniques have been proposed to address this

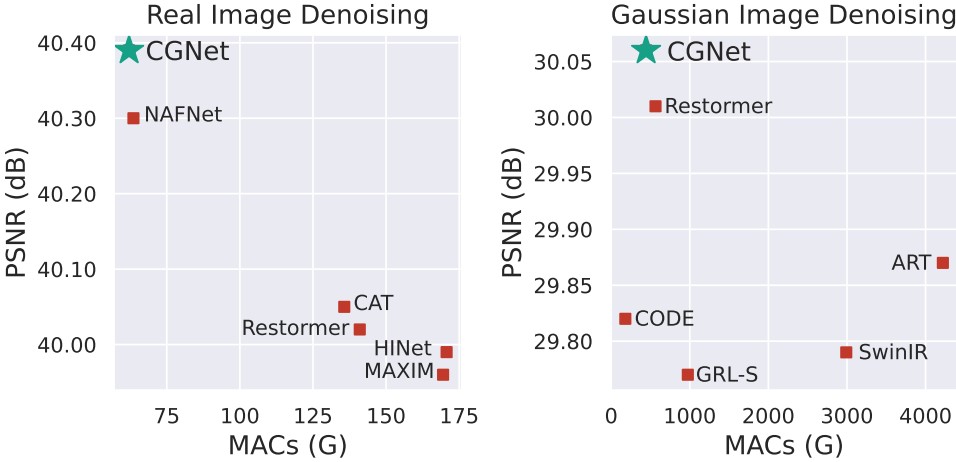

Figure 1: **Computational Efficiency vs Performance.** Left: PSNR vs. MACs (G) comparison on SIDD real image denoising. Right: PSNR vs. MACs (G) comparison on Gaussian image denoising tested on Kodak24 dataset with noise level $\sigma = 50$. Our model achieves state-of-the-art results and is computationally efficient.

challenge, including local attention (Wang et al., 2022b; Liang et al., 2021), which applies self-attention to smaller input patches instead of the entire input. Channel attention introduced by Restormer (Zamir et al., 2022) is another method that applies the attention mechanism to the channel dimension rather than the spatial dimension. Even though these methods have demonstrated improved computational efficiency, they do not fully capture long-range spatial dependencies. Building upon efficient attention mechanisms, various architectures have emerged, combining existing mechanisms or introducing novel attention methods to learn global context. Nonetheless, these approaches, including (Li et al., 2023b; Zhang et al., 2022; Chen et al., 2022b; Zhao et al., 2023), still require significant computational resources.

In this paper, we address the substantial computational overhead associated with learning global dependencies. We propose CascadedGaze Network (or CGNet), a fully convolutional encoder-decoder based restoration architecture, which uses Global Context Extractor (GCE) module to effectively capture the global context without relying on a self-attention mechanism, thus achieving both state-of-the-art performance and computational efficiency simultaneously in image restoration tasks. The name "CascadedGaze" reflects the cascading convolutional layers within the GCE. CGNet draws inspiration from recent work Metaformer (Yu et al., 2022) which challenges the prevailing belief that attention-based token mixer modules are essential for the competence of Transformers. Metaformer demonstrated that these attention-based modules can be replaced with simpler components while maintaining impressive performance.

We empirically demonstrate the efficacy of CGNet and the GCE when applied to image restoration tasks. We observe competitive performance on a range of benchmark datasets while maintaining a lower computational complexity and run-time compared to previous methods (Figure 1). In real image denoising (SIDD dataset), our method pushes the performance boundary further by surpassing the previous best-reported results. Further, on synthetic image denoising and single image motion deblurring (GoPro dataset), our method matches the state-of-the-art performance whilst being lower on MACs (G). These results emphasize the effectiveness of the proposed approach across various restoration tasks.

## 2 Related Work

The problem of image restoration is well-studied in computer vision literature (Fattal, 2007; HeK & SUNJ, 2011; Kopf et al., 2008; Michaeli & Irani, 2013), with competitions and challenges organized around designing methods across various restoration domains (Ignatov & Timofte, 2019; Abdelhamed et al., 2019; 2020; Li

et al., 2023c). In recent times, learnable neural network based approaches outperform the more traditional restoration methods (Chen et al., 2022a; Zhang et al., 2023; Chen et al., 2021; Zamir et al., 2021) even without any prior assumptions on the degradation process. Since these learnable approaches are data-driven, the availability of large-scale benchmark datasets allows these methods to estimate the distribution of degraded images empirically. This gain in performance is afforded by several stacked convolutional layers that downsample and upsample the feature maps throughout the network. Furthermore, most of these networks are constructed in U-Net (Ronneberger et al., 2015) fashion, where stacked convolutional layers form a U-shaped architecture with skip-connections providing the necessary signal over a longer range.

**Transformers in Restoration** Transformers have seen a significant surge in their usage across the suite of computer vision tasks, including image recognition, segmentation, and object detection (Dosovitskiy et al., 2020; Ramachandran et al., 2019; Touvron et al., 2021; Yuan et al., 2021; Liu et al., 2021; Carion et al., 2020); albeit they were originally designed for natural language tasks (Vaswani et al., 2017). Vision Transformers decompose images into sequences of patches and learn their relationships, demonstrating remarkable capabilities to handle long-range dependencies and adapt to diverse input content relying solely on self-attention to learn input and output representations. They have also been applied to low-level vision tasks like super-resolution, image colorization, denoising, and deraining (Zamir et al., 2022; Wang et al., 2022b; Liang et al., 2021; Tu et al., 2022; Li et al., 2023b; Zhao et al., 2023). Unlike high-level tasks, pixel-level challenges necessitate manipulating individual pixels or small pixel groups in an image to enhance or restore specific details. Although these architectures can learn long-term dependencies between sequences, the computational intractability hinders their realization and adoption in resource-constrained applications (Han et al., 2022; Lin et al., 2022). Specifically, the complexity increases quadratically with an increase in the input size.

**Efficient Transformers** Recent approaches seek alternative strategies that reduce complexity while ensuring the generation of high-resolution outputs (Liu et al., 2022a; Hatamizadeh et al., 2023; Liu et al., 2022c; Tang et al., 2022). One such approach is locality-constrained self-attention in Swin Transformer design (Liu et al., 2021). However, since self-attention is applied locally, the context aggregation is restricted to local neighborhoods. Some methods like CAT (Chen et al., 2022b) try to address the locality issue by using rectangle-window self-attention which utilizes horizontal and vertical rectangle-window attention to expand the attention area. A recent work, ART (Zhang et al., 2022), focused on combining sparse and dense attention, wherein the sparse attention module provides a wider receptive field and dense attention functions in a more local neighborhood. Low-rank factorization and approximation methods are two other efficient techniques employed to reduce the computational complexity of self-attention in Transformers (Wang et al., 2020; Xiong et al., 2021; Lu et al., 2021; Ma et al., 2021). However, these methods can lead to loss of information, are sensitive to hyper-parameters, and are potentially task-dependent.

**Fully Convolutional Methods in Restoration** Prior to the surge of Transformers, restoration methods utilized convolutional neural networks in their design (Tu et al., 2022; Zamir et al., 2021; Zhang et al., 2020b; Zamir et al., 2020). HINet (Chen et al., 2021), a multi-stage convolutional method, introduced the half-instance normalization block for image restoration. This was opposed to the batch normalization given the high variance between patches of images, and difference in training and testing settings, a normal practice in low-vision tasks such as restoration. SPAIR (Purohit et al., 2021) designed distortion-guided networks consisting of two main components: a network to identify the degraded pixels, and a restoration network to restore the degraded pixels. Building on Restormer's (Zamir et al., 2022) computational savings by introducing channel attention instead of spatial attention and prioritizing simplicity in design, NAFNet (Chen et al., 2022a) proposed a simplified version of channel attention, achieving state-of-the-art performance while being much more computationally efficient. For a more detailed survey on deep learning based restoration methods, we refer the reader to the recent survey work (Su et al., 2022a).

## 3 Methodology

We aim to develop a module capable of efficiently learning local and global information from the input data. To this end, we propose a cascaded fully convolutional module that progressively captures this information. It serves as a low-cost alternative for the attention mechanism. We further introduce the Range Fuser module

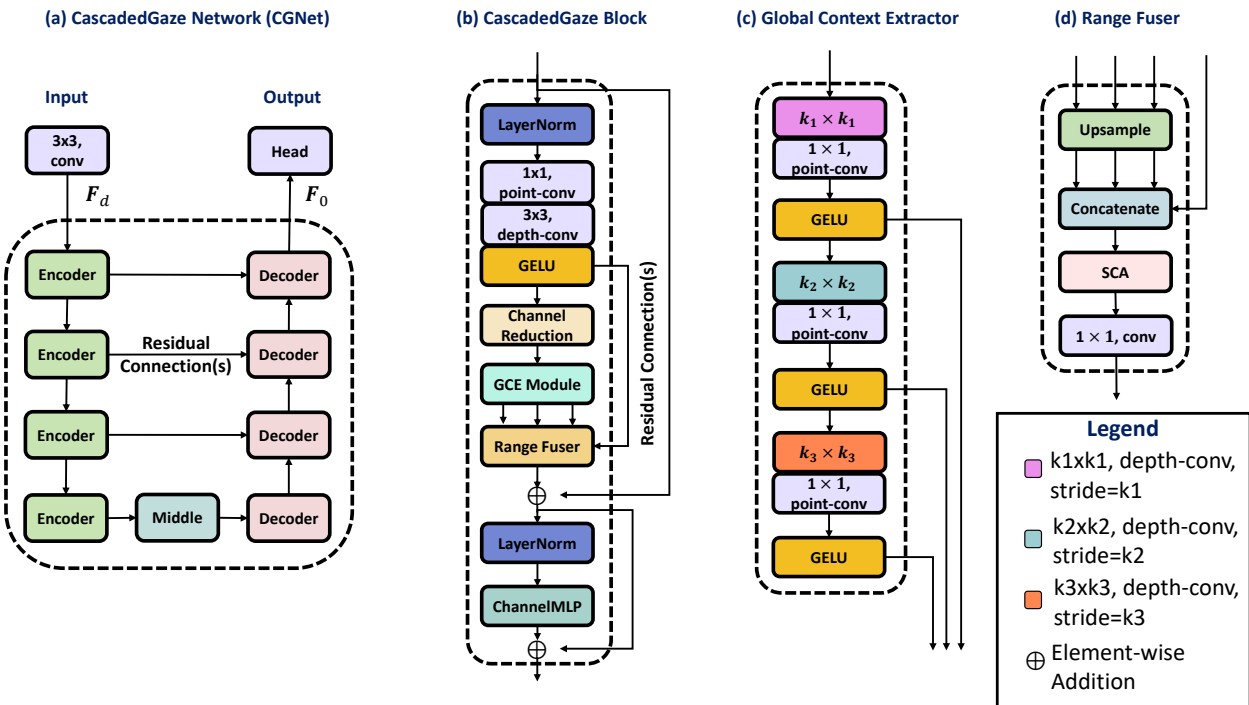

Figure 2: **Architecture Diagram.** (a) Illustration of the overall architecture of CascadedGaze network (CGNet). Each encoder layer comprises $N_g\times$ CascadedGaze blocks. (b) The CascadedGaze blocks are composed of (c) GCE module and (d) Range Fuser. GCE Module has three depthwise convolutions, followed by pointwise convolutions and GELU.

in order to aggregate the learned local and global context. Both of these modules are coupled with the restoration architecture, which we refer to as CascadedGaze Network (CGNet). In this section, we discuss the overall architecture of the proposed approach, followed by the two proposed modules, namely (a) Global Context Extractor (GCE) and (b) Range Fuser. Finally, we go through the details of construction steps to keep the architectural construction computationally tractable and efficient.

### 3.1 Overall Pipeline

We adopt the widely acknowledged U-shaped Net architecture, composed of several encoder-decoder blocks, which has emerged as a standard in image restoration tasks (Elad et al., 2023). We follow the supervised setting wherein the dataset $\boldsymbol{D}$ is realized by pairs of degraded and ground-truth (degradation-free) images $\boldsymbol{D} = \{(\hat{\boldsymbol{I}}_0, \boldsymbol{I}_0), (\hat{\boldsymbol{I}}_1, \boldsymbol{I}_1), ...., (\hat{\boldsymbol{I}}_n, \boldsymbol{I}_n)\}$ where $n$ is the total number of images, while $\hat{I}_i$, and $I_i$ denote the $i$th degraded and ground-truth images respectively.

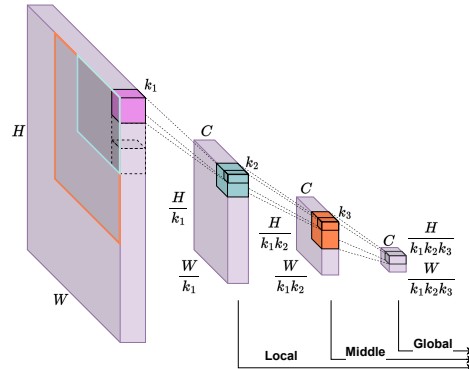

Figure 3: **GCE module.** We visualize the depthwise separable convolution layers to elucidate the capturing of context at different levels. The spatial range of each convolution is depicted in the input feature block with their corresponding colors.

Consider a degraded input image $\hat{\boldsymbol{I}} \in \mathbb{R}^{H \times W \times 3}$, where $H$, and $W$ denote the height, and width respectively (or spatial dimension of the image) and 3 denotes the number of channels. Input Image is first fed to the convolutional layer that transforms the image into a feature map $\mathcal{F}_0 \in \mathbb{R}^{H \times W \times C}$. This feature map then passes through four encoder-decoder stages. At each encoder stage, the input resolution is halved, and the number of channels is doubled. The

spatial dimension of the feature map is at the lowest in the middle block. Each encoder block is composed of $N_g\times$ CascadedGaze (CG) blocks. Since the U-Net structure is symmetric, each decoder block operates on input from the previous block and its corresponding encoder block through a skip connection. In each decoder stage, a pixel shuffling operation progressively restores the feature map's original resolution. The output from the last decoder block, $\mathcal{F}_d$, is then fed through the head of the network before the restored image $\boldsymbol{I_R} \in \mathbb{R}^{H\times W\times 3}$ is output. We defer the reader to Figure 2 for visualization of the entire architecture.

## 3.2 Global Context Extractor (GCE) Module

We draw insights from the work Metaformer (Yu et al., 2022) and find that it is possible to gain competitive performance by retaining the core structure of the Transformer, but replacing the self-attention mechanism with a more efficient alternative. We achieve the aforementioned goal by using convolutional layers with small kernel sizes as the building block of our Global Context Extractor (GCE) module to extract and aggregate features from a large area of the input feature map.

Each GCE module is composed of up to three convolution layers denoted by $l_1$, $l_2$, and $l_3$, respectively. For every convolution inside the GCE module, we set the stride to be equal to kernel size at each layer resulting in non-overlapping patches and subsequent reduction in the spatial dimension. The output spatial resolution of any convolution layer can be derived from $n_{\text{out}} = [\frac{n_{\text{in}}+2p-k}{s} + 1]$, where $p$ denotes padding, $s$ denotes stride, and $k$ denotes kernel size. In GCE, we set $s = k$, and $p = 0$, and re-write the above formula as $n_{\text{out}} = [\frac{n_{\text{in}}}{k}]$.

Consider the input feature map $\boldsymbol{F}_0^i \in \mathbb{R}^{H\times W\times C}$ that is fed to the $i$-th encoder. Let $G_j^i$ denote the $j$-th GCE module in $i$-th encoder. At the first convolution layer, $l_1$, has a kernel size of $k_1$, and will aggregate spatial features from a $k_1 \times k_1$ neighborhood of the input feature map. Let the output of $l_1$ be denoted by $\boldsymbol{A}^{\text{local}}$, then we can write it formally as

$$\boldsymbol{A}^{\text{local}} = G_j^i[l_1](\boldsymbol{F}_0^i) \in \mathbb{R}^{\frac{H}{k_1} \times \frac{W}{k_1} \times C}, \tag{1}$$

where $\frac{H}{k_1}$, and $\frac{W}{k_1}$, and $C$ denote the spatial dimension of the output feature map. Similarly, the second convolution layer, $l_2$, with a kernel size of $k_2$, will aggregate information from $k_2 \times k_2$ patches of summary tokens from the previous layer. Each of these summary tokens represents a $k_1 \times k_1$ area of the original input feature map, so the output of the second convolution can be described as aggregated information from a $k_1 k_2 \times k_1 k_2$ neighborhood of the input feature map. If the kernel size of the last convolution layer, $l_3$ is $k_3$, we can write a similar formal construction for $l_2$ and $l_3$.

$$\boldsymbol{A}^{\text{middle}} = G_j^i[l_2](\boldsymbol{A}^{\text{local}}) \in \mathbb{R}^{\frac{H}{k_1 \times k_2} \times \frac{W}{k_1 \times k_2} \times C}, \tag{2}$$

$$\boldsymbol{A}^{\text{global}} = G_j^i[l_3](\boldsymbol{A}^{\text{middle}}) \in \mathbb{R}^{\frac{H}{k_1 \times k_2 \times k_3} \times \frac{W}{k_1 \times k_2 \times k_3} \times C}, \tag{3}$$

where $\boldsymbol{A}^{\text{local}}$, $\boldsymbol{A}^{\text{middle}}$, and $\boldsymbol{A}^{\text{global}}$ denote local, middle, and global context, respectively.

**Comparison to Self-Attention**  Self-attention in ViT  (Dosovitskiy et al., 2020) functions on patches of images generated by splitting the image into fixed-sized pieces. Each patch, or its linear projection, is coupled with a 1D positional embedding indicating its position in the sequence. Self-attention then computes the attention score by attending to each sub-sequence (or patch) within the sequence (or image). In contrast, each subsequent layer in GCE operates on the *patches* generated by the layer preceding it. In Eq. 2, $\boldsymbol{A}^{\text{middle}}$ operates on the patches generated by the preceding layer, $\boldsymbol{A}^{\text{local}}$. Similarly, $\boldsymbol{A}^{\text{global}}$ operates on the patches generated by $\boldsymbol{A}^{\text{middle}}$ drawing parallels to self-attention.

## 3.3 Range Fuser

The extracted local and global features have different spatial sizes. To enable proper concatenation, we employ upsampling with nearest-neighbor interpolation to match the spatial dimensions. This is a non-learnable layer and hence does not affect the model size. We concatenate the upsampled feature maps along the channel dimension and obtain features with the original spatial dimensions but with inflated channels.

We recognize the varying importance of channels and draw inspiration from (Chen et al., 2022a) by employing Simple Channel Attention (SCA) to re-weight each channel. This approach allows for accentuating important channels while suppressing less informative ones, resulting in a more refined and focused representation of the aggregated features.

Further, we employ a single pointwise convolution to streamline the representation further and reduce the channel dimension to the input size. This yields a compact, refined input representation that seamlessly incorporates local and global information. Combining SCA and pointwise convolution ensures that our model retains the essential details while suppressing noise and improving performance and robustness.

### 3.4 On Computationally Efficient Construction

To further reduce the computational overhead, we merge similar channels by element-wise summation before feeding them to GCE. We explore two channel-merging options in this regard.

- **DynamicMerge:** We employ a dynamic channel merging technique by leveraging the token merging approach introduced by (Bolya et al., 2023) for merging similar tokens within Transformers. This adaptation relies on a selected similarity metric, such as Mean Absolute Error or correlation, to assess the channel similarity and facilitate the merging process. The similarity may be calculated among the channels themselves or the kernel weights of the depthwise convolution layer corresponding to each channel.

- **StaticMerge:** As opposed to a dynamic merging strategy, we also explore statically merging based on a fixed index. We achieve this by merging even channels with odd channels.

We ablate each method and find that static merging of channels (StaticMerge) performs the best in our case. This is preferable given that there is a constant computational cost to the operation; more discussion on the ablation experiments to follow.

### 3.5 Comparison to NAF Block

We compare the proposed CacadedGaze (CG) block and draw parallels with the previously introduced NAF block in (Chen et al., 2022a). The key difference lies in the CG block's Global Context Extractor (GCE), which utilizes cascading receptive fields for a compact understanding of input features at both local and global scales. Unlike the NAF block that relies on Simple Channel Attention (SCA) to re-weight channel-wise feature maps, CG block learns a local and global compact aggregated representation without the computational overhead of Transformer-like architectures.

## 4 Results

We evaluate CGNet on benchmark datasets for three image restoration tasks (a) real image denoising, (b) Gaussian image denoising, and (c) single image motion deblurring. We discuss these restoration tasks, and datasets, and then describe our experimental setup, hyperparameters, and training protocol, and results.

Figure 4: **Block Comparison.** CacadedGaze Block and NAF Block comparison diagram.

### 4.1 Datasets

For image denoising, we train our models on both synthetic benchmark datasets (Gaussian image denoising) and the real-world noise dataset (real image denoising). The Smartphone Image Denoising Dataset (SIDD) (Abdelhamed et al., 2018) is a real-world noise dataset composed of images captured from different smartphones under various lighting and ISO conditions, inducing a variety of noise levels in the images. The

Table 1: **Scores on Gaussian Image Denoising task.** We report PSNR scores along with MACs (G) and inference time (milliseconds) (denoted Inf.) calculated for an image size of $512 \times 512 \times 3$ one a single NVIDIA Tesla v100 PCIe 32 GB GPU. Our method is lower in MACs and is faster than previous methods while remaining comparable, if not better. Notably, our model outperforms Restormer, which has the closest MACs and inference time to us, across all test datasets except for McMaster. The best results are highlighted in red, while the second bests are in blue. ⋆We do not highlight ART (Zhang et al., 2022) due to significant differences in MACs.

| Method | MACs ↓ (G) | Inf. ↓ (ms) | CBSD68 (2001) | | | Kodak24 (1999) | | | McMaster (2011) | | | Urban100 (2015) | | |
|---|---|---|---|---|---|---|---|---|---|---|---|---|---|---|
| | | | $\sigma = 15$ | $\sigma = 25$ | $\sigma = 50$ | $\sigma = 15$ | $\sigma = 25$ | $\sigma = 50$ | $\sigma = 15$ | $\sigma = 25$ | $\sigma = 50$ | $\sigma = 15$ | $\sigma = 25$ | $\sigma = 50$ |
| SwinIR (2021) | 2991 | 1850 | 34.42 | 31.78 | 28.56 | 35.34 | 32.89 | 29.79 | 35.61 | 33.20 | 30.22 | 35.13 | 32.90 | 29.82 |
| Restormer (2022) | 564 | 350 | 34.40 | 31.79 | 28.60 | 35.47 | 33.04 | 30.01 | 35.61 | 33.34 | 30.30 | 35.13 | 32.96 | 30.02 |
| GRL-S (2023b) | 975 | 680 | 34.36 | 31.72 | 28.51 | 35.32 | 32.88 | 29.77 | 35.32 | 33.29 | 30.18 | 35.24 | 33.07 | 30.09 |
| ART⋆ (2022) | 4220 | OOM | 34.46 | 31.84 | 28.63 | 35.39 | 32.95 | 29.87 | 35.68 | 33.41 | 30.31 | 35.29 | 33.14 | 30.19 |
| CODE (2023) | 180 | 600 | 34.33 | 31.69 | 28.47 | 35.32 | 32.88 | 29.82 | 35.38 | 33.11 | 30.03 | – | – | – |
| **CGNet** (Ours) | **444** | **215** | 34.41 | 31.79 | 28.60 | 35.52 | 33.07 | 30.06 | 35.58 | 33.28 | 30.22 | 35.18 | 32.98 | 30.07 |

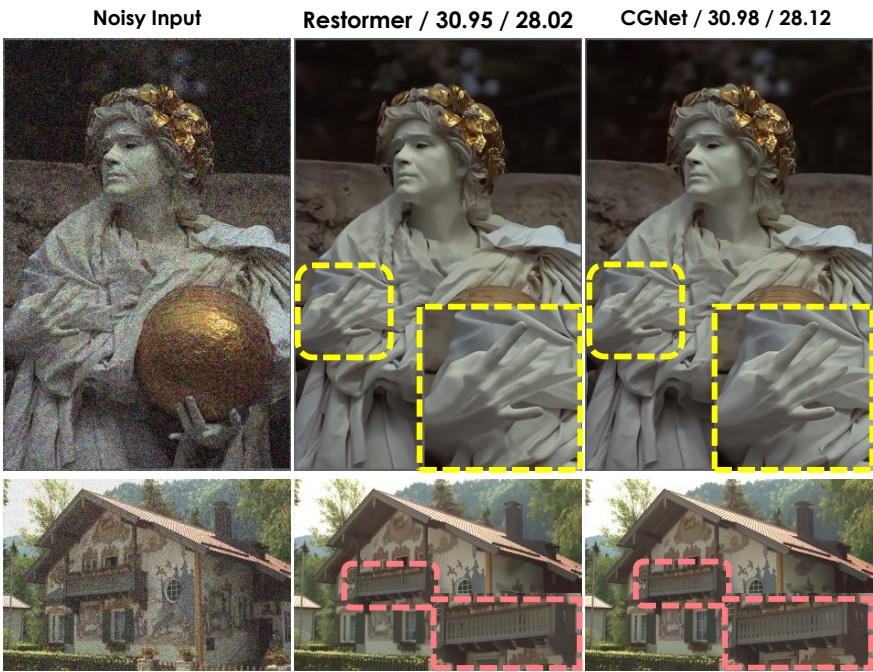

Figure 5: **Qualitative Comparison on Gaussian Denoising.** Visual results on Gaussian image denoising on Kodak24 (Franzen, 1999) dataset. We compare with Restormer (Zamir et al., 2022), the best method in the literature on the dataset. Our method, CGNet, restores finer details and pleasing outputs. The corresponding PSNR scores for each image are mentioned at the top of the figure.

synthetic benchmark datasets are generated with additive white Gaussian noise on BSD68 (Martin et al., 2001), Urban100 (Huang et al., 2015), Kodak24 (Franzen, 1999) and McMaster (Zhang et al., 2011). For image motion deblurring, we employ the GoPro dataset (Nah et al., 2017) as the training data. The GoPro dataset contains dynamic motion blurred scenes captured from a consumer-grade camera. In all cases, we adopt the standard data preprocessing pipeline following (Chen et al., 2021; 2022a; Zamir et al., 2022)

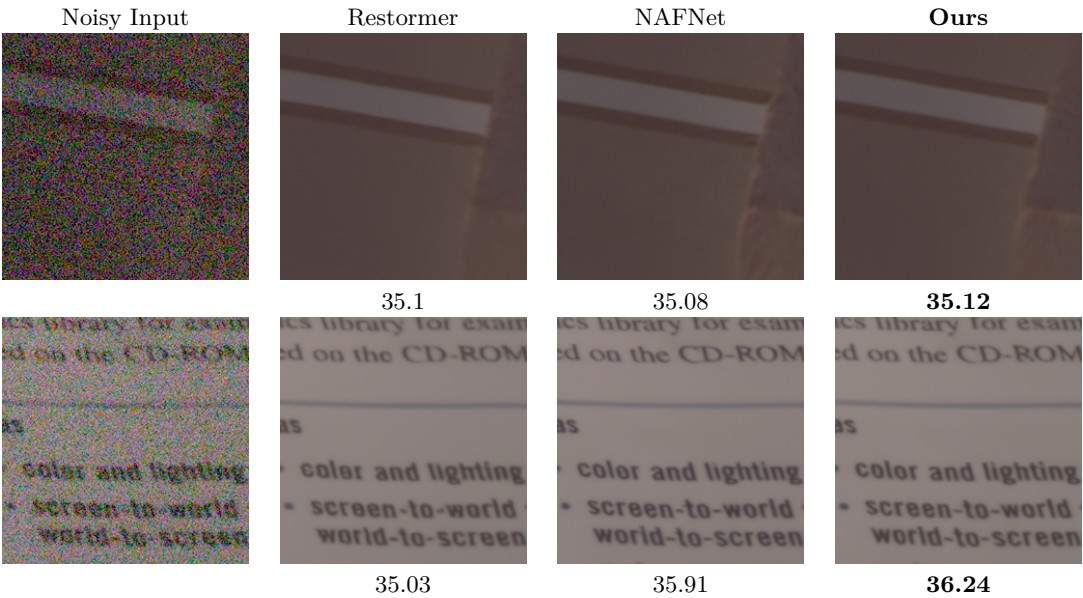

Figure 6: **Qualitative Comparison on Real Image Denoising.** Denoising results on validation images from SIDD dataset (Abdelhamed et al., 2018). CGNet (Ours) restores visually pleasing images in a variety of scenes and objects; additionally, the PSNR scores quantitatively confirm CGNet's performance boost in these images.

Table 2: **Scores on Image Denoising on SIDD dataset.** CGNet achieves state-of-the-art results on the SIDD dataset while being faster with a lower MACs. The inference time is calculated on a single NVIDIA Tesla v100 PCIe 32 GB GPU, and the MACs is calculated for an image size of $256 \times 256 \times 3$. Note that we do not report inference time for methods scoring much lower PSNR on the task. The best results are highlighted in red, while the second best in blue.

| Smartphone Image Denoising Dataset (SIDD) | | | | |
|---|---|---|---|---|
| **Method** | **MACs ↓ (G)** | **Inference ↓ Time (ms)** | **PSNR ↑** | **SSIM ↑** |
| **MPRNet** (Zamir et al., 2021) | 588 | – | 39.17 | 0.958 |
| **CycleISP** (Zamir et al., 2020) | 189.5 | – | 39.52 | 0.957 |
| **HINet** (Chen et al., 2021) | 170.7 | – | 39.99 | 0.960 |
| **MAXIM** (Tu et al., 2022) | 169.5 | – | 39.96 | 0.958 |
| **CAT** (Chen et al., 2022b) | 135.7 | 390 | 40.05 | 0.960 |
| **Restormer** (Zamir et al., 2022) | 141.0 | 102 | 40.02 | 0.960 |
| **NAFNet** (Chen et al., 2022a) | 63.6 | 53 | **40.30** | **0.962** |
| **CGNet** (Ours) | **62.1** | **52** | **40.39** | **0.964** |

## 4.2 Experimental Setup

CGNet for the denoising and deblurring tasks comprises a sequence of four encoder blocks, one middle block, followed by a sequence of four decoder blocks, with skip connections between corresponding encoder/decoder blocks. To reduce computational expenses, we strategically use CGE blocks in the encoder and simple NAF

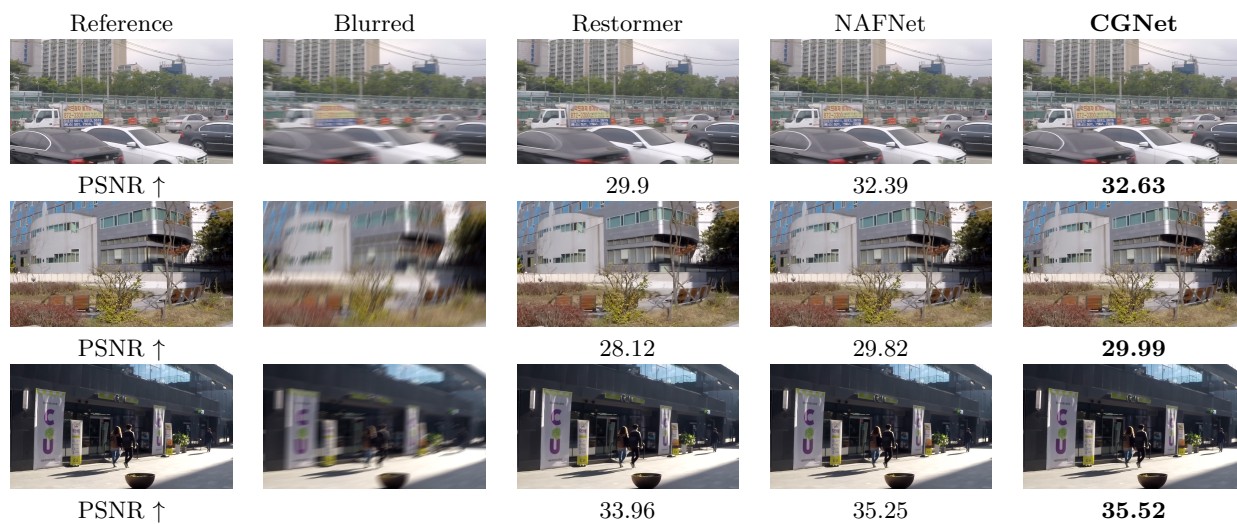

Figure 7: **Qualitative Comparison on Image Deblurring.** Visual results on Single Image Motion Deblurring on sample images from validation set of GoPro dataset Nah et al. (2017). CGNet (ours) results are much more closely aligned with the ground truth in terms of reconstruction, and are sharper.

blocks (Chen et al., 2022a) in other places which is discussed in detail in the ablation section. Below, we elaborate on the specifications of each model.

**Real Image Denoising** The encoder comprises 2, 2, 4, and 6 CascadedGaze blocks, respectively. The rest of the network is composed of NAF blocks with 10 at the middle layer and 2, 2, 2 and 2 for the four decoder blocks, respectively. We set the width of the network to 60. The restored image is taken from the head of the network, which is a convolutional layer applied to the output of the last decoder.

**Gaussian Image Denoising.** For a fair comparison with previous methods in the literature, we increase the size of our network – specifically, increasing the number of blocks and the width. The encoder has 4, 4, 6, and 8 blocks, the middle layer has 10 blocks at each stage, and the decoder has 2, 2, 2, and 4 blocks. We set the width of the network to 70.

**Image Deblurring.** The first three encoder blocks have 1 CascadedGaze block each, while the fourth encoder comprises 2 CascadedGaze blocks followed by 25 NAF blocks. The remaining middle and decoder blocks also comprise 1 NAFNet block each. We set the width of the network to 62 in this case. For the deblurring task, we follow the architectural modifications proposed in the work (Liu et al., 2022b). Unlike the single output in the denoising model, we modify the head of the network to accommodate for $K$ multiple outputs allowing the network to output multiple feasible solutions. We set the value $K = 4$ for all of these experiments. Since the models are trained on $256 \times 256$ patch sizes, testing on larger sizes degrades performance; therefore, we finetune the model on $384 \times 384$ patches for 2 more epochs following (Zamir et al., 2022); additionally, we use TLC as proposed by (Chu et al., 2022) for inference on image deblurring task.

**Shared Configuration.** We train the models in all tasks for $400K$ iterations, with AdamW as the optimizer ($\beta_1 = 0.9, \beta_2 = 0.9$), and minimize the negative PSNR loss function (i.e., maximize the PSNR). We use a cosine annealing scheduler that starts with the learning rate of $1e^{-3}$ and decays to $1e^{-7}$ throughout learning. All of our models are implemented in the PyTorch library, trained on 8 NVIDIA Tesla v100 PCIe 32 GB GPUs. For inference, we utilize a single GPU. During training for real denoising and motion deblurring experiments, we set the image patch size to $256 \times 256$. For Gaussian denoising, we follow (Zamir et al., 2022)'s progressive training configuration and start with the patch size of 160 and increase it to 192, 256, 320, and 384 during training. The reported results are averaged over three runs. We compute Peak Signal-to-Noise

Table 3: **Scores on Singe Image Motion Deblurring on GoPro dataset.** Our method scores the highest PSNR on the task, achieving state-of-the-art results. The best results are highlighted in red, while the second bests are in blue.

| GoPro Motion Deblurring | | | | | |
|---|---|---|---|---|---|
| **Method** | **SRN** (Tao et al., 2018) | **DBGAN** (Zhang et al., 2020a) | **SPAIR** Purohit et al. (2021) | **MPRNet** (Zamir et al., 2021) | **HINet** (Chen et al., 2021) | **HI-Diff** (Chen et al., 2024) |
| **PSNR** | 30.26 | 31.10 | 32.06 | 32.66 | 32.77 | 33.33 |
| **SSIM** | 0.934 | 0.942 | 0.953 | 0.959 | 0.959 | 0.964 |
| **Method** | **MAXIM** (Tu et al., 2022) | **Restormer** (Zamir et al., 2022) | **NAFNet** (Chen et al., 2022a) | **NAFNet MH-C** (Liu et al., 2022b) | **DiffIR** (Xia et al., 2023) | **CGNet** (Ours) |
| **PSNR** | 32.86 | 32.92 | 33.71 | 33.75 | 33.20 | 33.77 |
| **SSIM** | 0.961 | 0.961 | 0.967 | 0.967 | 0.963 | 0.968 |

Ratio (PSNR) metric and Structural Similarity Index (SSIM) in line with the standard evaluation protocol followed by literature on image restoration (Chen et al., 2022a; Purohit et al., 2021; Zamir et al., 2022).

## 4.3 Results Discussion

**Real Image Denoising.** We perform experiments on the Smartphone Image Denoising Dataset (SIDD) (Abdelhamed et al., 2018) as part of the real-world denoising experiments. Table 2 compares CGNet with previously published methods in the literature. Our proposed approach archives 0.09 dB gain over the previous best method NAFNet (Zamir et al., 2022). We provide visual results on sample images from the SIDD dataset in Figure 6; our method restores results more faithfully and closer to the ground truth.

**Gaussian Image Denoising.** We present results of CGNet on Gaussian image denoising on four datasets with three different noise levels ($\sigma = 15, 25, 50$) in Table 1. As the spatial size of images in the test datasets is larger than 512, the reported MACs (G) values are calculated for an image size of $512 \times 512$. Our method is comparable to current state-of-the-art methods, pushing the boundary on a few datasets while being significantly faster in inference time, and lower on MACs (G). We beat Restormer (Zamir et al., 2022) in all datasets except McMaster. Even though we have reported ART (Zhang et al., 2022), we note that CGNet is not comparable as ART's MACs (G) is $10\times$ larger. Also, it is computationally intractable on a limited budget, given that we observe an Out of Memory (OOM) error when running inference on ART. We present a few visual results in Figure 5 on the Kodak24 dataset.

**Single Image Motion Deblurring.** Table 3 lists the results of our approach on single image motion deblurring task on the GoPro dataset (Nah et al., 2017). Our model gains $+0.06$ and $+0.02$ dB in PSNR compared to NAFNet and NAFNet multi-head methods, showing the effectiveness of our method in different restoration tasks. Furthermore, visual results on images from the GoPro dataset are also provided in Figure 7.

## 4.4 Visualizing GCE Module

We visualize the GCE module, mainly looking at how local and global layers learn input context. Figure 8 plots the activations of $\boldsymbol{A}^{\text{local}}$ and $\boldsymbol{A}^{\text{global}}$, recall Eq. 1 and Eq. 3, from $G_1^2$ i.e. the first GCE module of second encoder block. The layer operating on the local context learns structure local to foreground objects occupying considerable pixel space in the image (for example, the cars), while the global context is much broader with considerably activated objects present even in the background (for example, trees and sidewalk). For each image, the local context acts like an edge detector learning low-level features local to the objects. Notice how objects much further away in the distance are void of sharp edges. On the other hand, the global context learns higher abstractions of the image than low-level features. Such analysis of neuron activations to understand context is well explored in interpretability literature, both in language processing (Sajjad et al., 2022), and vision (Zeiler & Fergus, 2014).

| Input | **Local Context** | **Global Context** |
|---|---|---|

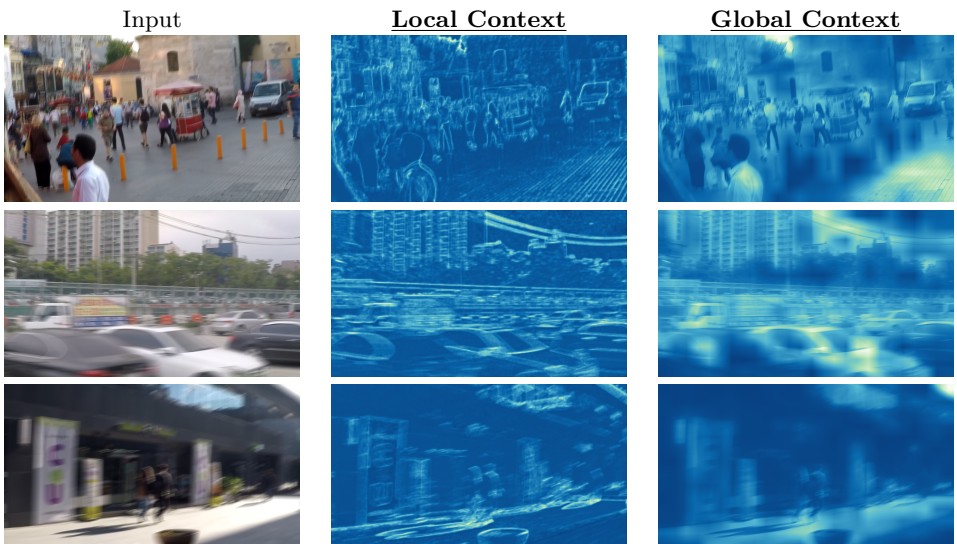

Figure 8: **GCE Module Visualized.** Visualization of the local and global context taken from the outputs of the Global Context Extractor (GCE) module. Results visualized on images taken from validation set of GoPro dataset (Nah et al., 2017). The local context is adept at learning local structure and features – edges, whereas the global context is extracting high-level features and shapes.

## 4.5 Ablation Study

We ablate the proposed CascadedGazeNet to understand what components necessitate efficiency and performance gains. All experiments are conducted on real-image denoising task using a smaller variant of our model with a singular block at each level of the architecture, and a width of 8. Our smaller models operate within a computational budget of approximately 0.5 MACs (G), and are trained for a total of $200K$ iterations while the remaining settings are the same as those of the main model. In all the cases, the combinations we adopt for the CascadedGazeNet are in **bold**.

**Channel Merging Method.** We employ a merging algorithm before using the GCE module to reduce its computational overhead further. As shown in Table 7, our ablation study showed that merging channels based on a fixed index, referred to as StaticMerge, during both training and inference outperforms dynamically merging similar channels (referred to as DynamicMerge). The simplicity of the method makes it easier for the model to learn, as it does not have to adapt to a different combination of channels for each batch of data.

**Kernel Sizes.** The choice of kernel sizes significantly impacts the performance of a CNN. In general, it is better to use smaller kernel sizes at the beginning of the GCE and then use larger kernel sizes for later convolutional layers. Smaller kernel sizes allow the GCE to extract more detailed information from the input image. Then, larger kernels at the end of the GCE aggregate these details to capture more global information. We follow this intuition and ablate two kernel choices for each layer in the GCE module. We aim to understand whether smaller kernel to larger kernel sizes is better for design or vice-versa. Table 5 shows our experiments' results on kernel size choices. Our results show that the best choice is to utilize a smaller to larger kernel size design, where the initial layer extracts local, while the last layer learns global context.

**Global Context Extractor Module Placement.** The GCE module is a resource-intensive component, and incorporating it extensively throughout the network is impractical. This stems from the trade-off between performance enhancements and computational costs, necessitating careful equilibrium. Intuitively, since GCE extracts both local and global information, it is best suited for the encoder part. This helps the

Table 4: **GCE Block Place Study.** We ablate the placement of GCE blocks throughout the network. Enc refers to Encoder blocks, while Mid refers to Middle blocks, and Dec refers to Decoder blocks.

| GCE Location | | | PSNR | MACs (G) | Params (M) |
|---|---|---|---|---|---|
| Enc | Mid | Dec | | | |
| ✓ | ✗ | ✗ | **39.32** | 0.446 | 0.406 |
| ✓ | ✓ | ✗ | 39.32 | 0.460 | 0.737 |
| ✓ | ✗ | ✓ | 39.32 | 0.506 | 0.517 |
| ✓ | ✓ | ✓ | 39.33 | 0.520 | 0.848 |

Table 5: **Kernel Size Study.** Comparison of different kernel sizes for the GCE module in the architecture. We ablate two combinations to determine the optimal employment of larger kernels at the initial and final stages.

| Kernel Sizes | PSNR | MACs (G) | Params (M) |
|---|---|---|---|
| $[5, 3, 3]$ | 39.25 | 0.442 | 0.408 |
| $[3, 3, 5]$ | **39.32** | 0.446 | 0.406 |

Table 6: **Comparison of Different Convolutional Layers.** We mainly ablate the order of pointwise (PW) and depthwise convolutions (DW) and compare these combinations with the standard convolutional layer.

| Convolution Type | PSNR | MACs (G) | Params (M) |
|---|---|---|---|
| Standard | 39.33 | 0.480 | 0.498 |
| PW+DW | 39.32 | 0.464 | 0.406 |
| DW+PW | **39.32** | 0.446 | 0.406 |

model utilize information to capture fine-grained non-corrupted information. However, we ablate the GCE placement, cumulatively increasing the GCE blocks throughout the network. The results are summarized in Table 4. Our experiments back up the idea that placing the GCE module in the encoder blocks yields the best balance between performance and computational efficiency.

**Channel Expansion before GCE.** We investigate the effects of channel expansion at the beginning of the CascadedGaze block using a point-wise convolution operation. Specifically, we try expanding by ×2, keeping it as is, and expanding by ×2 while utilizing channel merging (StaticMerge) to reduce the number of channels by half. In agreement with intuition, we find that expanding the channels by ×2, and performing reduction by channel merging (StaticMerge) works the best while maintaining a balance between the MACs (G) and PSNR score. The complete analysis is shown in Table 8.

**Convolutional Layer Type.** The type of convolutional layers used in the GCE module significantly impacts our model's size and computational efficiency. Therefore, we ablate the choice by considering three options: standard convolution, pointwise convolution + depthwise convolution (PW+DW), and depthwise convolution + pointwise convolution (DW+PW). As shown in Table 6, using depthwise convolution to reduce the spatial dimension and then applying the pointwise convolution significantly makes our model smaller while maintaining competitive performance.

Table 7: **Comparison of Channel Merging.** We report comparison on two channel merging methods. DynamicMerge has different strategies, whereas StaticMerge is a fixed merging method, as discussed before.

| Method | Strategy | PSNR | Inference Time (ms) |
|---|---|---|---|
| StaticMerge | Fixed | **39.32** | 14.5 |
| DynamicMerge | Channel Cosine Similarity | 39.26 | 16 |
| | Kernel Cosine Similarity | 39.29 | 14.5 |
| | Kernel MAE | 39.28 | 14.5 |

Table 8: **Comparison of Channel Expansion.** We compare channel expansion before passing to GCE module. When channels are reduced to half $\frac{C}{2}$, we use the StaticMerge technique to achieve the desired reduction.

| Expansion Factor | Channels | PSNR | MACs (G) | Params (M) |
|---|---|---|---|---|
| ×2 | $2 \times C$ | 39.33 | 0.507 | 0.569 |
| ×1 | $C$ | 39.28 | 0.401 | 0.355 |
| ×2 | $(2 \times C)/2$ | **39.32** | 0.446 | 0.406 |

## 5 Conclusion

We introduced a method to learn the local and global context for image restoration tasks in a computationally efficient manner. Inspired by the self-attention mechanism in Transformers, we proposed a module termed Global Context Extractor (GCE), for fully convolutional architectures. We constructed a restoration architecture, termed CascadedGaze Network (CGNet), utilizing the introduced GCE module and empirically verified the effectiveness in terms of overall performance and computational tractability. We hope that our work will spur interest in efficient architecture construction to learn the global context for various low-level vision tasks.

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

Table 9: **Generalization Experiments.** On several denoising, and deblurring datasets (synthetic and real-world), we report performance on the generalization experiments. The MACs(G) are computed on an input size of $(512 \times 512 \times 3)$.

| | | | Generalization Experiment(s) | | | | | | | |
|---|---|---|---|---|---|---|---|---|---|---|
| | | | Real-World Denosing | | Synthetic Blur | | Real-World Blur | | | |
| Method | MACs (G) | Params | DND (2017) | | GoPro (2017) | | RealBlurR (2020) | | RealBlurJ (2020) | |
| | | | PSNR | SSIM | PSNR | SSIM | PSNR | SSIM | PSNR | SSIM |
| NAFNet (2022a) | 254.37G | 115.98M | 38.41 | 0.943 | 33.71 | 0.967 | 36.07 | 0.954 | 28.18 | 0.848 |
| CGNet (Ours) | 248.48G | 119.22M | 39.41 (+1.0) dB | 0.950 (+0.007) | 33.77 (+0.06) dB | 0.968 (+0.001) | 36.04 (−0.03) dB | 0.954 (−−) | 28.28 (+0.1) dB | 0.853 (+0.005) |

Table 10: **Synthetic Gaussian Denoising Dataset.** Experiments comparing NAFNet, and CGNet (ours) on four synthetic Gaussian denoising datasets with noise level $\sigma = 25$. ∗ We trained NAFNet given the lack of open-source official results from the authors.

| Synthetic Gaussian Denoising | | | | |
|---|---|---|---|---|
| $\sigma = 25$ | CBSD68 (2001) | Kodak (1999) | McMaster (2011) | Urban100 (2015) |
| NAFNet* (2022a) | 31.75 | 33.01 | 33.26 | 32.83 |
| CGNet (Ours) | 31.79 (+0.04) dB | 33.07 (+0.06) dB | 33.28 (+0.02) dB | 32.98 (+0.15) dB |

# Appendix

# A    Generalization Experiments

In this section, we discuss the generalization experiments on denoising and deblurring tasks. We test our proposed method on four new datasets and compare it with the methods in the literature. More specifically, we test our image denoising method, trained on SIDD dataset (Abdelhamed et al., 2018), on the Darmstadt Noise Dataset (DND) (Plotz & Roth, 2017). Further, we test the single image deblurring method, trained on synthetic blur datasets GoPro (Nah et al., 2017), on another synthetic blur dataset HIDE (Shen et al., 2019), and two real-world blur datasets, RealBlurR (Rim et al., 2020) and RealBlurJ (Rim et al., 2020). Our method is a non-transformer architecture and is comparable in parameters and MACs (G) to the previously introduced NAFNet (Chen et al., 2022a); therefore, we emphasize the comparison with the aforementioned method. We use the official models trained on SIDD and GoPro, which were published by the authors[1], and only run inference for comparison.

## A.1    Synthetic Image Denoising

To compare NAFNet on synthetic Gaussian image denoising task with noise level $\sigma = 25$, we train the method since there is no official trained model made available by the authors. We increase the model capacity and size to match MACs(G) of CGNet for fair comparison. The results, PSNR scores, are reported in Table 10; our method scores higher on the metric across all four datasets.

## A.2    Real Image Denoising

Darmstadt Noise Dataset (DND) (Plotz & Roth, 2017) consists of 50 pairs of real-noise and its ground-truth images captured with several different consumer-grade cameras. The reference image is taken with the base ISO level, while the noisy image is captured with a higher ISO. The dataset is prepared following a series of processing steps to handle camera shift alignment, exposure time adjustment, and intensity scaling.

---

[1] https://github.com/megvii-research/NAFNet

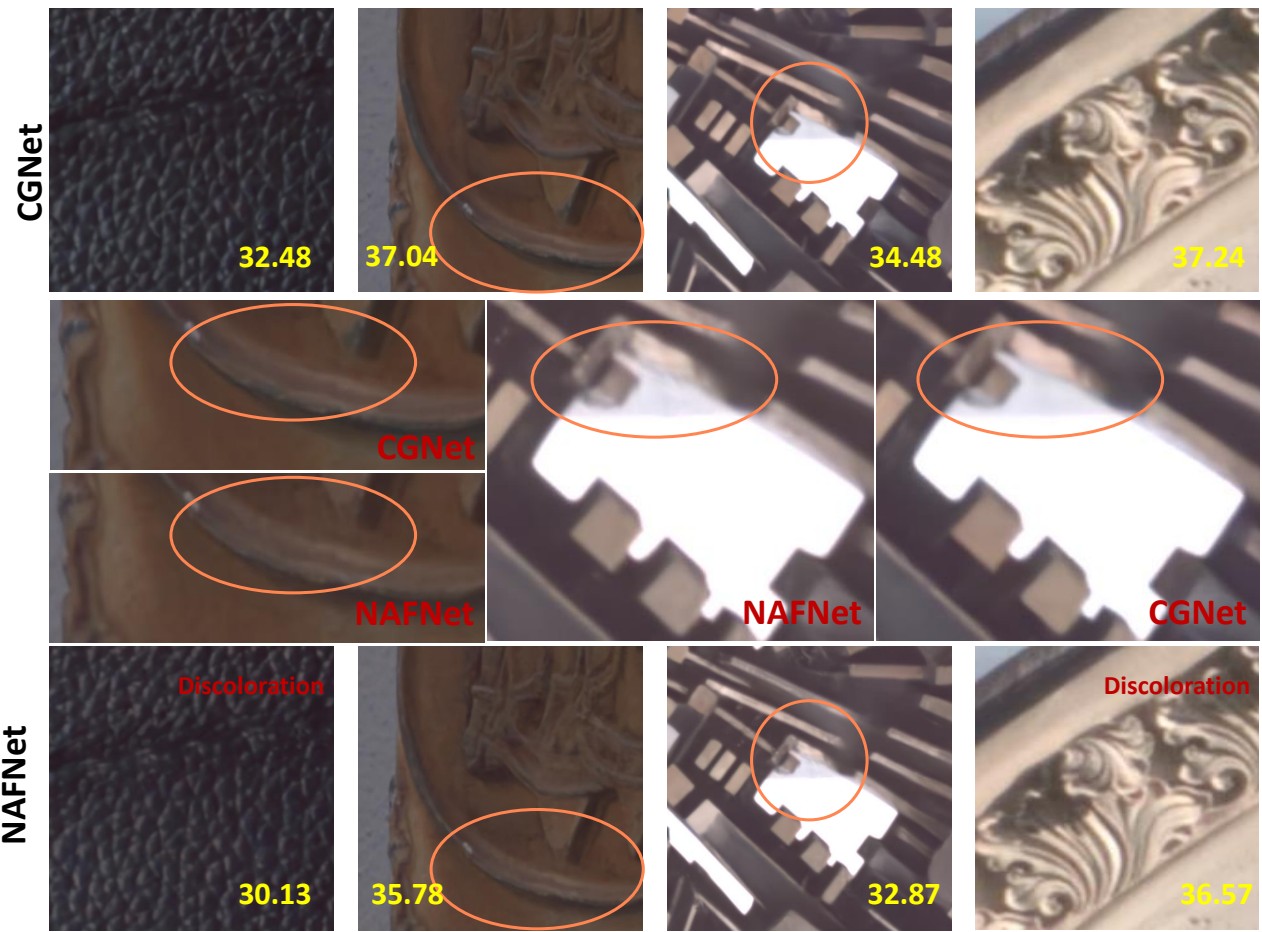

Figure 9: **Qualitative Comparison on Real Image Denoising.** Visual Results of Real-World Denoising on the DND dataset. The zoomed-in regions are provided for visualization, while the PSNR scores are written on top of the restored frames.

It is common practice in image restoration literature to evaluate methods trained on the SIDD dataset on the DND benchmark test since both are real-world noised datasets, albeit with different capturing devices (smartphones and consumer-grade cameras, respectively). We compare our proposed CGNet method with NAFNet given that both are non-Transformer architectures and are comparable in MACs (G). We observe that our method scores +1.0 dB higher in PSNR on the task compared to NAFNet in Table 9, and the restored images are artifacts-free and faithful to the ground-truth, see Figure 9.

### A.3 Real-World Image Deblurring

On real-world image deblurring task, we compare both GoPro dataset trained models on two different real-world blurry datasets: RealBlurR, and RealBlurJ (Rim et al., 2020). The datasets are constructed with an acquisition system designed to capture aligned pairs of blurred and sharp images, followed by post-processing techniques to construct high-quality ground-truth images. On RealBlurJ, our proposed method (CGNet) scores higher both on PSNR and SSIM metrics, while is comparable to NAFNet on the Real-BlurR dataset. In conclusion, we find that CGNet generalizes better than NAFNet on unseen degradation (denoising, deblurring–both synthetic and real-world).

Table 11: **Comparison on Single Image Motion Deblurring with Diffusion Methods.** We report results on the single image motion deblurring task on the GoPro dataset Nah et al. (2017) with recent diffusion-based restoration models.

| Methods | DiffIR (2023) | InD (2023) | HI-Diff (2024) | DvSR (2022) | DvSR-SA (2022) | Swintormer (2024) | CGNet (Ours) |
|---------|---------------|------------|----------------|-------------|----------------|-------------------|--------------|
| **PSNR** | 33.20 | 31.49 | 33.33 | 31.66 | 33.23 | 33.38 | **33.77** |
| **SSIM** | 0.963 | 0.946 | 0.964 | 0.948 | 0.963 | 0.965 | **0.968** |

## B   Restoration Problems and Diffusion Models

Diffusion models have revolutionized the generative modeling landscape, and now have started to find their adoption, although gradual, in the restoration community. In supervised restoration tasks, agreement of the restored results with the ground truth is of high importance to avoid unwanted artifacts, and hallucinated details, a problem inherent to naive diffusion models (Chen et al., 2024). However, more recently, diffusion models have shown competitive performance on several restoration tasks, such as image deblurring, image in-painting, image super-resolution, and image dehazing (Kawar et al., 2022; Wang et al., 2022a; Chung et al., 2022; Delbracio & Milanfar, 2023). Generally, diffusion models require several steps (due to their iterative nature) to produce a high-fidelity output and hence, encounter computational intractability issues. Recent works on diffusion models for restoration, such as (Xia et al., 2023; Chen et al., 2024; Whang et al., 2022), now consider computational efficiency to be an important design decision of the architecture formalism. We refer the reader to the recent survey work on diffusion methods for image restoration (Li et al., 2023a) for an in-depth literature summary.

In Table 11, we compare our proposed approach with recent diffusion models based image deblurring architectures on the GoPro dataset (Nah et al., 2017). Notably, diffusion models are known to score on the lower end on distortion metrics such as PSNR, and SSIM due to the prevalence of undesired artifacts in the restored results, or misaligned generations (Chen et al., 2024; Whang et al., 2022).

