# OpenReview forum: "CascadedGaze: Efficiency in Global Context Extraction for Image Restoration"
_TMLR — Accepted by TMLR_

### Review · Reviewer_dgkX · 2024-02-29

**Summary Of Contributions:**

This paper proposes a new network architecture CascadedGaze Network (CGNet), which is a convolutional network that captures the global features using a Global Context Extractor (GCE). The idea is to achieve the success of transformers but be more efficient by proposing a component that does a similar job of self-attention but is convolutional. The effectiveness of the proposed CGNet architecture was verified on the SIDD dataset, the Gaussian image denoising dataset, and the motion deblurring dataset GoPro.

**Audience:**

Yes

**Claims And Evidence:**

Yes

**Requested Changes:**

More comparison with NAFNet or exploring ways to combine GCE with NAFNet.

**Strengths And Weaknesses:**

Strengths:
1. Proposed a novel convolutional architecture that works as well as transformers but is more efficient.

2. The new component of the network Global Context Extractor (GCE) is well motivated.

3. The empirical result demonstrates the advantage of the proposed CGNet network.

Weaknesses:
1. The performance is very close to NAFNet, especially on the SIDD) and GoPro datasets.

---

### Review · Reviewer_DDmx · 2024-03-06

**Summary Of Contributions:**

The authors propose two architectural modifications to make neural networks for image restoration more computationally efficient without sacrificing performance. First, a Global Context Extractor, inspired by Metaformer, that captures global information without self-attention. Second, a Range Fuser that combines channels corresponding to different scales.

**Audience:**

Yes

**Claims And Evidence:**

No

**Requested Changes:**

1. Tone down the claims on state-of-the-art performance.
2. Relate or, better, compare with recent diffusion models for image restoration.

**Strengths And Weaknesses:**

Overall, the paper is very well-written and, as far as I can tell, the experimental setup follows current best practices.
The design choices are clearly motivated and I appreciate in particular that Section 4.4 (Figure 7) also verifies that they agree with experiments.

While the efficiency improvement in terms of MACs is evident, I would however argue that the PSNR improvements are insignificant. In general, I think the abstract and introduction make too strong claims about being state-of-the-art, while the experimental section gives a more balanced account, e.g. _"Our method is comparable to current state-of-the-art methods, pushing the boundary on a few datasets while being significantly faster in inference time, and lower on MACs (G)."_

Related to these SotA claims, why is there no reference to the NTIRE 2023 Image Denoising challenge? I'm also missing an account of recent advances in diffusion models for image restoration, see e.g. this survey: https://arxiv.org/abs/2308.09388

---

### Review · Reviewer_w7yt · 2024-03-12

**Summary Of Contributions:**

This article proposes a CascadedGaze Network (CGNet), an encoder-decoder architecture that employs Global Context Extractor (GCE) to capture global information for image restoration. Compared with the traditional convolution network, it considers the global context of the image. Compared with the traditional transformer network, it reduces the computational overhead. Judging from the experimental results, there is a certain performance improvement in RSNR and MACs.

**Audience:**

Yes

**Claims And Evidence:**

Yes

**Requested Changes:**

1.	Figure texts need to be enlarged.
2.	It is necessary to propose more novel network architecture or algorithm theory rather than simple model splicing.
3.	More favorable experimental results compared with NAFNet need to be added to prove the value of this work.

**Strengths And Weaknesses:**

Strengths:
1.	The relevant work statement is more comprehensive, and the language expression is smooth.
2.	A new CascadedGaze Network is proposed for image restoration tasks, and good results are obtained on PSNR and MACs.
Weaknesses:
1.	The overall innovation of the paper is limited. It only integrates U-Net, cascaded pyramid convolution, and NAFNet's Simple Channel Attention (SCA), and almost no original content exists.
2.	The fonts in Figures 2, 3, and Table 1 are too small and can hardly be read on printed paper.
3.	The performance improvement of the experimental part compared to NAFNet is very small, which is not enough to support the advantages of the entire paper.

---

> ### Author Response · Authors · 2024-03-14
> **Response to Reviewer w7yt: more results with NAFNet, and commentary on architecture novelty**
>
> We thank the reviewer for reading our manuscript, and providing constructive feedback. The requested changes by the reviewer are addressed as follows.
>
> **Figure texts need to be enlarged.**
>
> We have fixed the Figure 2, and 3 fonts, and have also fixed the fonts in Table 1.
>
> **More favorable experimental results compared with NAFNet need to be added to prove the value of this work.**
>
> We have conducted further experiments to compare our method with NAFNet on Darmstadt Noise Dataset (DND) [real-world image denoising], four synthetic denoising datasets, and RealBlur [real-world image deblurring] datasets. All of the new results, along with the visual quality analysis are reported in the appendix A in red color, specifically in Tables 9, and 10, and Figure 9. We summarize the results in the following points.
>
> 1. For the real-world denoising task on the Darmstadt Noise Dataset (DND), CGNet outperformed NAFNet by +1.0dB over NAFNet, achieving a score of 39.41 dB. The visual comparison is reported in Figure 8.
>
> 2. For synthetic image denoising, we train NAFNet on noise level of σ = 25, while adapting the NAFNet to match the MACs (G) of CGNet for fair comparison. Our proposed method outperformed or matched the performance of NAFNet across all the four evaluated datasets. Notably, on the Urban100 dataset, CGNet achieved a PSNR of 32.98, surpassing NAFNet's 32.83.
>
> 3. On real-world image deblurring, CGNet exhibited competitive performance, particularly notable on the RealBlurJ dataset with a PSNR of 28.28 dB, exceeding NAFNet by +0.1 dB, while staying comparable on RealBlurR.
>
> **It is necessary to propose more novel network architecture or algorithm theory rather than simple model splicing.**
>
> Several of the restoration architectures/methods are mainly deployed on edge devices and are not run in the cloud. Therefore, computational cost is an important design decision.
> While attention is superior in its ability to learn context over a longer range, it is also expensive. There have been several variants of attention introduced in the literature (Restormer [1], GRL [2], CODE [3], ART [4]), however, the only economical choice is NAFNet which is a non-transformer architecture and relies on Simple Channel Attention (SCA), a simplified variant of channel attention.
>
> As compared to NAFNet, our work is not incremental in design or performance. In general, our method performs better than or competitively with NAFNet. However, in certain cases (such as DND, SIDD, Urban100) our performance is much better. More importantly, our method restores the quality faithfully, without introducing any artifacts or textures (Figure 9). We argue that the reason is NAFNet's SCA and the limited capacity of SCA, by design, to understand longer spatial context. In contrast to this, we proposed the CascadedGaze Block which contains the Global Context Extractor (GCE) module functioning to learn and aggregate context at different spatial resolutions. Lastly, it is important to note that it is significantly challenging to beat NAFNet given it was introduced in 2022, and, to the best of our knowledge, it has not seen a competitor on a similar MACs(G), and parameters budget, but with better performance (Figure 1).
>
> Admittedly, on the operator level, we are not introducing new operators. Rather the focus is to construct an optimal overall architecture to utilize global spatial information while achieving computational tractability. To this end, we motivate our design choices and provide experimental evidence on a variety of tasks to back up the claims (both quantitative and qualitative). With this work, we are trying to be accurate, and robust in experimentation in order to set a new Pareto frontier. Lastly, we think that this work will be interesting to the restoration community at large wherein the goal is to maintain some order of hardware-friendliness in architecture design and avoid any unwanted artifacts in the restored results.
>
> [1] https://arxiv.org/abs/2111.09881 \
> [2] https://arxiv.org/abs/2303.00748 \
> [3] https://openaccess.thecvf.com/content/CVPR2023/papers/Zhao_Comprehensive_and_Delicate_An_Efficient_Transformer_for_Image_Restoration_CVPR_2023_paper.pdf \
> [4] https://arxiv.org/abs/2210.01427

---

### Author Response · Authors · 2024-04-18
**Summary of Changes**

We would like to thank all the reviewers for their insightful comments and suggestions. We believe that we have addressed all of the comments, and have incorporated all the suggestions as well. We summarize the changes made to the manuscript as follows.

**Figure/Text Adjustments:** We made adjustments to Figures 2, and 3 and also fixed the fonts in Table 1, as per the suggestion of Reviewer w7yt. Further, we made adjustments to the text to tone down the introduction, and the abstract as per the comments of Reviewer DDmx.

**More Experimental Results Compared with NAFNet:** We added more results compared to NAFNet in Tables 9, and 10, and also added qualitative analysis in Figure 9 as per the suggestion of Reviewers dgkX, and w7yt.

**Comparison with Diffusion Models:** We also added a comparison with recent diffusion models on a similar task to us (such as GoPro deblurring) in Table 11, and provided some commentary on the results and trends in diffusion methods for image restoration based on the suggestion of Reviewer DDmx.

We are grateful for the comments and would be open to further discussion/comments on the manuscript.

---

### Decision · Action_Editor_hetc · 2024-04-26

**Recommendation:** Accept as is

**Comment:**

3 reviewers are initially skeptical w.r.t. the significance compared to the baseline NAFNet. After the authors' revision and rebuttal, those concerns were largely lifted. All reviewers were convinced that CGNet improves over NAFNet and recommended acceptance. Editor hence follow the reviewers' unanimous recommendations to recommend acceptance of the paper.

**Audience:**

The paper provides a new approach to image restoration hence that community would be interested. Its reduction of the computational overhead over transformers may have some general interest to a wider audience.

**Claims And Evidence:**

This paper proposed CascadeGaze Network (CGNet), an encoder-decoder architecture that employs Global Context Extractor (GCE) to capture global information for image restoration. Compared with the traditional convolution network, it considers the global context of the image. Compared with the traditional transformer network, it reduces the computational overhead. Judging from the experimental results, there is a certain performance improvement in RSNR and MACs.

3 reviewers are initially skeptical w.r.t. the significance compared to the baseline NAFNet. After the authors' revision and rebuttal, those concerns were largely lifted. All reviewers were convinced that CGNet improves over NAFNet.